# ES-GGT: Efficient Submap-based Visual Geometry Grounded Transformer with Spatial Memory Alignment

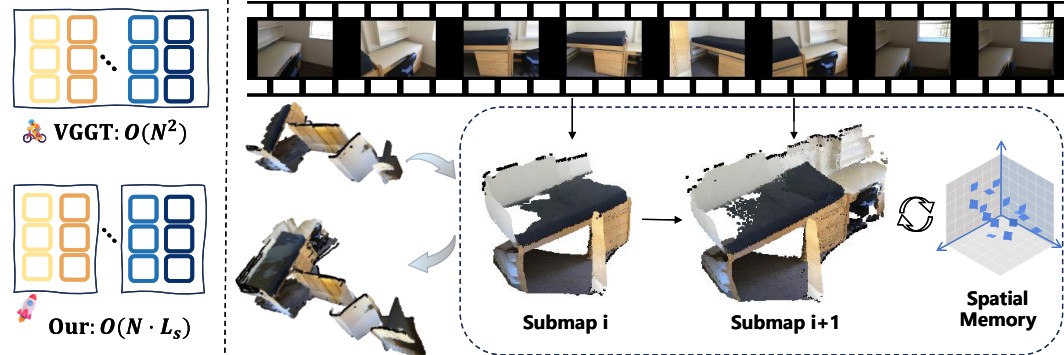

Figure 1: **Left:** The complexity of the algorithm is depicted, where $L_s$ represents the number of images per submaps and $N$ denotes the total number of input images. **Right:** The algorithmic diagram illustrates the process, where multiple submaps are first reconstructed in a streaming manner into a group, and different groups are then sequentially merged to produce the final prediction.

## Abstract

Foundation models have recently emerged as powerful tools in 3D vision, greatly advancing the field of 3D perception. However, improving computational efficiency while maintaining consistency in long sequences remains a key challenge in computer vision. We present ES-GGT, an efficient method for streaming scene reconstruction built on VGGT, a state-of-the-art feed-forward visual geometry model. We align submaps in a streaming manner using a hierarchical, local-to-global strategy. At the local level, we perform fine-grained alignment of their scales and coordinate systems by streaming low-level information, thereby reducing computational complexity while maintaining memory cost and performance comparable to simultaneous input of all submaps. For global level, we integrate high-level spatial memory with a tri-perspective view (TPV) representation that extends the bird's-eye view (BEV) with two orthogonal planes. We then generate a 15-degrees-of-freedom homography transformation matrix to achieve global alignment. We significantly improved inference speed and efficiently handled long sequence inputs. Code available at: https://anonymous.4open.science/r/ES-GGT-4386.

## 1 Introduction

Dense 3D scene reconstruction from monocular RGB images is a fundamental problem in computer vision, with wide applications in robotics, augmented reality, and autonomous navigation (Liu et al. (2025); Raychaudhuri et al. (2024); Khazatsky et al. (2024)). Recent advances in feed-forward neural reconstruction models have significantly improved the quality and efficiency of 3D perception. Notably, methods such as DUSt3R (Wang et al. (2024)), MASt3R (Leroy et al. (2024)), and VGGT (Wang et al. (2025)) have demonstrated the ability to predict dense geometry and camera

poses directly from images, bypassing traditional multi-stage pipelines like Structure-from-Motion (SfM) (Frahm et al. (2010); Liu et al. (2024a); Gu et al. (2020)) and Multi-View Stereo (MVS) (Furukawa & Hernández (2015); Huang et al. (2018); Galliani et al. (2015); Wang et al. (2021) ). These models leverage powerful architectures and large-scale training data to achieve impressive reconstruction quality.

Dispite their success, extending these feed-forward methods to long video sequences remains a critical challenge. Most existing approaches are limited by GPU memory constraints and a computational complexity that scales quadratically ($\mathbf{O}(N^2)$) with the number of input frames. For instance, VGGT (Wang et al. (2025)), while capable of processing arbitrary numbers of views, suffers from a quadratic scaling of computational cost due to its global attention mechanism. This limits its applicability in streaming or large-scale reconstruction scenarios (Wang* et al. (2025)). To address this, recent works like VGGT-SLAM (Wang et al. (2025)) propose dividing the input into submaps or sliding windows and aligning them incrementally. While these methods improve scalability, they often rely on strong assumptions about camera calibration or scene structure, and may struggle with drift accumulation or misalignment in challenging environments.

In this paper, we present ES-GGT, an Efficient Submap-based Visual Geometry Grounded Transformer (Vaswani et al. (2017)) designed for scalable and consistent 3D reconstruction from long RGB sequences. As illustrated in Figure 1, our approach processes long image sequences in a streaming manner, dramatically reducing computational complexity from

Built upon the VGGT architecture, ES-GGT introduces a hierarchical alignment strategy that processes input images in streaming submaps, significantly reducing computational complexity from $\mathbf{O}(N^2)$ to $\mathbf{O}(N \cdot L_S)$, where $N$ is the number of input images and $L_s$ is the image unmber of each submap. At the local level, we enforce fine-grained consistency across overlapping frames within each group of submaps using a novel cross-submap alignment mechanism. At the global level, we maintain a spatial memory representation using a Tri-Perspective View (TPV) (Huang et al. (2023)) and estimate a 15-degree-of-freedom homography transformation (Hartley & Zisserman (2003)) to align submaps in a globally consistent coordinate system.

Unlike VGGT-SLAM , which aligns submaps using SL(4) transformations and assumes projective ambiguity, ES-GGT avoids costly global optimization by integrating spatial memory directly into the feed-forward process. Compared to SLAM3R (Liu et al. (2024b)), which focuses on real-time registration without explicit camera estimation, our method retains the geometric interpretability of VGGT while improving efficiency and long-term consistency. Extensive experiments on 7-Scenes dataset (Schonberger & Frahm (2016)) demonstrate that ES-GGT achieves superior reconstruction accuracy and completeness.

Our contributions can be summarized as follows:

- Propose ES-GGT, a submap-based transformer architecture build on VGGT that enables efficient 3D reconstruction from monocular RGB images. And significantly reduce computational complexity.

- Introduce a hierarchical alignment strategy that integrates intra-group fine-grained consistency with inter-group global alignment, leveraging spatial memory and homography estimation.

- Demonstrate that ES-GGT surpasses existing methods in both reconstruction quality and computational efficiency. When processing more than 100 input frames, our method achieves over 3× speedup compared to VGGT. On the 7-Scenes dataset, our reconstruction results achieve state-of-the-art performance.

## 2 RELATED WORKS

### 2.1 FEED-FORWARD 3D SCENE RECONSTRUCTION

Feed-forward neural methods have recently achieved remarkable progress in dense 3D reconstruction (Duisterhof et al. (2025b); Murai et al. (2024); Zhang et al. (2024); Szymanowicz et al. (2025); Li et al. (2025b); Xiao et al. (2025); Li et al. (2025a)). Departing from traditional optimization-heavy pipelines such as Structure-from-Motion (SfM) and Multi-View Stereo (MVS) (Schönberger

& Frahm (2016); Schönberger et al. (2016); Agarwal et al. (2011); Nistér (2004); Hartley (1997); Liu et al. (2024a); Yao et al. (2018); Mouragnon et al. (2006); He et al. (2024); Gu et al. (2020); Ding et al. (2022); Schönberger et al. (2016)), feed-forward models now enable direct inference of 3D structure and camera poses from RGB inputs. Pioneering works such as DUSt3R (Wang et al. (2024)) demonstrated that a network can directly regress dense pointmaps from uncalibrated image pairs. This paradigm has inspired numerous follow-up works. To extend this capability to video sequences, methods like Spann3R (Wang & Agapito (2024)) and Cut3R (Wang* et al. (2025)) introduced recurrent mechanisms and persistent state tokens to process frames incrementally. SLAM3R (Liu et al. (2024b)) further developed this concept by using a sliding window to reconstruct local geometry and then registering these clips into a global scene representation. While these incremental methods improve efficiency, they are susceptible to cumulative drift over long sequences. Other works like Pow3R (Jang et al. (2025)) focus on improving reconstruction quality by incorporating priors like known camera parameters or sparse depth maps at test time. The core ideas from these models have also been extended to other 3D representations, such as directly outputting Gaussian Splatting parameters (Smart et al. (2024); Sun et al. (2025)). Our work, in contrast, addresses the scalability and drift challenges through a novel hierarchical alignment strategy that does not rely on additional priors.

## 2.2 Transformer Architectures for Multi-View Geometry

Recent advances in transformer-based architectures have significantly reshaped the landscape of multi-view 3D geometry estimation (Wang et al. (2025); Xiao et al. (2025); Zhang et al. (2025); Duisterhof et al. (2025a); Keetha et al. (2025); Wang et al. (2025); Khafizov et al. (2025)). VGGT (Wang et al. (2025)) introduces a unified transformer architecture that jointly estimates camera parameters, depth maps, and dense point clouds in a single forward pass. By alternating between frame-wise and global self-attention layers, VGGT captures long-range spatial dependencies across views. However, the global attention mechanism that underpins VGGT's strong performance is also its primary limitation. The model's computational and memory requirements scale quadratically with the number of input frames, rendering it impractical for long video sequences or real-time applications. FastVGGT (Shen et al. (2025)) attempts to accelerate inference by merging redundant tokens. Fast3R (Yang et al. (2025)) designs global fusion transformers to process a larger number of views simultaneously, but this still faces scalability challenges with very long contexts. Our work, ES-GGT, directly tackles this challenge by partitioning the input sequence into manageable submaps, thus breaking the quadratic dependency.

## 2.3 Submap-based Reconstruction

To scale powerful feed-forward models like VGGT to arbitrary-length sequences, a "divide-and-merge" strategy has become the prevailing approach. This involves breaking the sequence into smaller, overlapping submaps, processing each independently, and then aligning them into a globally consistent model (Deng et al. (2025); Maggio et al. (2025)). Recent SLAM systems built on feed-forward backbones have adopted this strategy, but differ significantly in their alignment philosophies. VGGT-SLAM (Maggio et al. (2025)) extends VGGT by first generates submaps using VGGT and then addresses the 15-DoF projective ambiguity inherent in reconstructions from uncalibrated cameras. It formulates a factor graph optimization that operates directly on the SL(4) manifold to estimate the projective transformations (homographies) between submaps. MASt3R-SLAM (Murai et al. (2024)) builds upon the two-view MASt3R model and employs a backend with Sim(3) pose graph optimization to ensure global consistency. While effective, these methods bifurcate reconstruction and alignment into distinct, often computationally intensive, steps. SLAM3R (Liu et al. (2024b)) takes a different, fully end-to-end learning approach. It avoids explicit camera pose estimation by using a Local-to-World (L2W) network to directly register new pointmaps into a global frame. This is guided by a memory reservoir of previously observed scene frames. These approaches, however, leave two critical challenges unaddressed: (i) how to ensure fine-grained geometric consistency across multiple submaps within a local window in a purely feed-forward manner, and (ii) how to perform robust global alignment without resorting to a separate, costly optimization loop. ES-GGT bridges this gap. Our hierarchical alignment strategy integrates an intra-group feature propagation mechanism for local consistency with a learnable, TPV-based spatial memory for global alignment. This allows ES-GGT to achieve scalable, consistent reconstruction in a single forward pass while retaining the valuable geometric interpretability of the VGGT framework.

## 3 REVIEW: VGGT

VGGT (Wang et al. (2025)) is a feed-forward transformer that processes a set of $N$ RGB images, $\{I_i \in \mathbb{R}^{3 \times H \times W}\}_{i=1}^N$, and generates a complete 3D scene description for each frame in a single forward pass. For each input image $I_i$, the network estimates camera parameters $g_i$, consisting of a quaternion, translation vector, and field of view, along with a dense depth map $D_i$, a viewpoint-invariant point map $P_i$ expressed in the coordinate frame of the first camera, and $C$-dimensional tracking features $T_i$ (Karaev et al. (2024a;b)).

$$f_{\text{vggt}} : \mathcal{I} \to \mathcal{O}, \ \mathcal{I} = \{I_i \in \mathbb{R}^{3 \times H \times W}\}_{i=1}^N$$
$$\mathcal{O} = \{(g_i, D_i, P_i, T_i)\}_{i=1}^N$$

The backbone is a 24-layer Vision Transformer whose tokens are produced by a frozen DI-NOv2 (Oquab et al. (2023)) patchifier. To reason efficiently across many views, the transformer alternates between two self-attention modes: a frame attention layer that updates tokens within each individual image, and a global attention layer that exchanges information across all frames. The output tokens are subsequently processed by a camera head to predict camera intrinsics and poses, or by Dense Prediction Transformer (DPT) heads (Ranftl et al. (2021)), which generate dense depth maps for each image, a dense point map, and per-pixel feature embeddings for point tracking. This architecture does not employ any cross-attention layers, only self-attention ones. Since the global attention layer in VGGT is designed to capture complex geometric relationships across all input frames, its computational complexity scales quadratically with the sequence length, which quickly emerges as a major performance bottleneck. To alleviate this issue, we partition the input into submaps, effectively reducing the computational overhead incurred by the global attention layer.

## 4 METHOD

We aim to design a network that, given an input sequence of $N$ images $I^{\text{input}} \in \mathbb{R}^{N \times H \times W \times 3}$, processes them in a submap manner, where each submap is represented as an image collection $I^{\text{s}} \in \mathbb{R}^{L_{\text{submap}} \times H \times W \times 3}$, and $L_{\text{submap}}$ corresponds to the number of images per submap. Each submap starts with $L_{\text{overlap}}$ overlapping frames inherited from its preceding submap, ensuring smooth temporal continuity. We treat $L_{\text{group}}$ as the number of submaps in a group, denoted as $I^{\text{g}} \in \mathbb{R}^{L_{\text{group}} \times L_{\text{submap}} \times H \times W \times 3}$, and process them jointly. For clarity of exposition, we assume throughout that the total sequence length $N$ is exactly divisible as $N = L_{\text{group}} \times L_{\text{submap}}$ Within each group, we stream low-level information across submaps to maintain high regional consistency in later inputs. Each group is processed to produce independent predictions that are subsequently aligned via a global spatial memory $\mathcal{M}$ to maintain global consistency between groups. By enforcing fine-grained, low-level alignment intra-group and promoting high-level alignment inter-group, our approach guarantees consistency among long-range submaps.

Overall, our alignment strategy proceeds in two stages: **intra-group alignment**, which refines the relative scales and coordinate frames among submaps within each group, and **inter-group alignment**, which integrates the already aligned grouped-submaps into a globally consistent representation.

### 4.1 INTRA-GROUP ALIGNMENT

Formally, the $j$-th group is constructed from a consecutive segment of the input submap as:

$$I_j^g = \{I_i^s | i \in [(j-1) \cdot L_{\text{submap}} + 1, j \cdot L_{\text{submap}}]\}.$$

Each submap $I^{\text{s}}$ serves as the atomic processing unit of the network. At each iteration, the network takes the i-th submap $I_i^{\text{s}}$ as input. Each image $img \in I_i^{\text{s}}$ is first patchified into a set of $K$ tokens using a DINO (Oquab et al. (2023)) encoder. The tokens from all frames within the submap are then concatenated and passed through the backbone, which alternates between frame attention and global attention layers.

We follow the original VGGT (Wang et al. (2025)) configuration and employ a backbone with 24 alternating layers of global and frame-wise attention. For each input $img$ in i-th submap, the

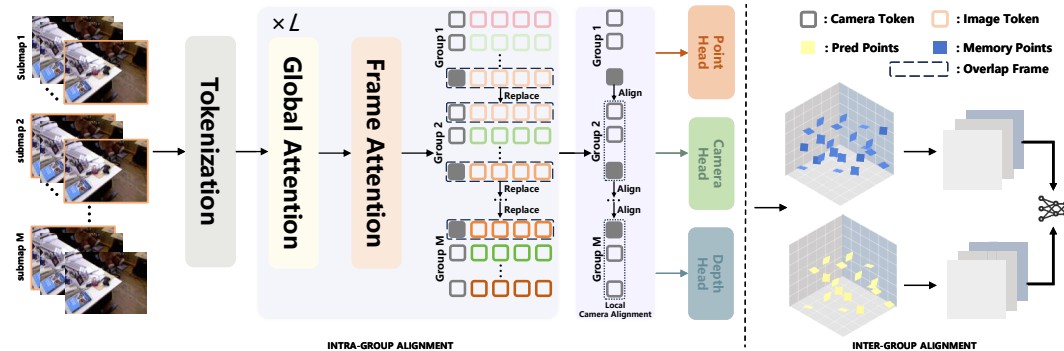

Figure 2: Overall pipeline of our method. Given an input sequence of $N$ images, we first divide it into $L_{\text{group}}$ groups, each group containing $L_{\text{submap}}$ images. Within each group, **intra-group alignment** propagates overlap-frame features and refines camera tokens to ensure local consistency across submaps. Subsequently, **inter-group alignment** integrates group-level predictions into a globally consistent point cloud via the global spatial memory $\mathcal{M}$. This two-stage alignment strategy enables both fine-grained local coherence and long-range global consistency in reconstruction.

backbone produces a feature representation $t^{\text{img}} \in \mathbb{R}^{24 \times 2 \times K \times C}$, where $K$ denotes the number of tokens and $C$ is the feature dimension.

To maintain temporal coherence between submaps, we introduce overlap frames $img_o$ that are shared between consecutive submaps. Simply re-encoding these frames, however, would limit the receptive field to the current submap. Instead, we propagate the feature representations $t^{img_o}$ from the last submap and substitute them for the corresponding feature in the current submap $I_i^s$. Importantly, this substitution is performed only in the global attention layers, allowing overlap tokens to carry forward contextual information and anchor the global computation across submaps.

For each $t^{img}$, the 0-th token corresponds to the camera token $c$, which encodes information related to the camera. In particular, the camera token of the first frame $I_0$ specifies the coordinate system for each prediction.

Since the prediction of camera parameters for image $img_i$ relies solely on its corresponding camera token $c_i$, we can interpret $c_i$ as encoding the camera coordinate system information of the submap. For all submaps within the same group, we expect their camera tokens to encode a consistent coordinate system. In particular, the camera tokens of overlap frames should remain as consistent as possible across consecutive submaps.

To enforce this consistency, we introduce a cross-submap regularization mechanism. Specifically, for each overlap frame $img_o$ shared between the $(i-1)$-th and $i$-th submaps, we compute a residual embedding by passing the difference of their camera tokens through a lightweight MLP:

$$r_0^{(i)} = MLP(c_0^{(i)} - c_0^{(i-1)}), i \in [2, L_{\text{group}}],$$

where $c^{(i)}o$ and $c^{(i-1)}o$ denote the camera tokens of the same overlap frame in consecutive submaps $I^si$ and $I^si - 1$.

We then aggregate these residuals across all overlap frames via average pooling, and use the resulting feature to refine the camera tokens of the entire $i$-th submap:

$$\tilde{c}_j^{(i)} = c_j^{(i)} + AvgPool(\{r_o^{(i)}\}_{o=1}^{L_{\text{overlap}}}), i \in [2, L_{\text{group}}], j \in [1, L_{\text{submap}}],$$

where $\tilde{c}_j^{(i)}$ denotes the updated camera token for the $j$-th image in submap $I_i^s$. This update allows overlap frames to propagate consistent camera information across submaps, while simultaneously aligning all camera tokens within the group to a shared coordinate system.

For each group, we jointly predict the camera parameters, point maps, and depth maps, all expressed in the coordinate frame of the first camera in the group.

## 4.2 INTER-GROUP ALIGNMENT

To achieve global consistency across groups, we maintain the global spatial memory $\mathcal{M}$ that stores high-level information from previously predicted points. Given a new group output $\mathcal{O}_i^g$, we employ the Sim(3) method to predict a rotation matrix, yielding an initially aligned point cloud. Subsequently, we query $\mathcal{M}$ to retrieve points $P_i^{\text{memory}}$ within the intersection of the predicted region $P_i^{\text{pred}}$ and the stored memory, determined by the Intersection over Union (IoU) which defines the region used for refinement.

We encode these 3D points with the Tri-Perspective View (TPV) comprising three orthogonal Bird's-Eye Views (BEVs). Formally, each BEV projection defines a point set

$$P^{\text{BEV}} = \{P_{u,v} \mid 1 \le u \le H_{\text{BEV}}, 1 \le v \le W_{\text{BEV}}\},$$

where $P_{u,v}$ denotes the set of projected points onto the $u$-$v$-th BEV plane.

After projection, we employ a Point-wise Feature Network (PFN) to extract local descriptors for each cell $P_{u,v}$, yielding a dense representation $\mathcal{F} \in \mathbb{R}^{3 \times H_{\text{BEV}} \times W_{\text{BEV}} \times C_{\text{BEV}}}$. We then fuse the memory feature $\mathcal{F}_i^{\text{memory}}$ and the predicted feature $\mathcal{F}_i^{\text{pred}}$ through a cross-attention module, producing an alignment representation $\mathcal{F}_i^{\text{align}}$. Finally, a lightweight regression head maps $\mathcal{F}_i^{\text{align}}$ to a 15-DoF correction matrix $\mathbf{T} \in \mathbb{R}^{4 \times 4}$ that enforces rigid alignment (with $\det(\mathbf{T}) = 1$), ensuring consistency between the predicted region and the spatial memory. The updated point set $\tilde{P}_i^{\text{pred}}$ is then merged into the global point cloud. To maintain memory efficiency, we apply voxel-grid downsampling.

## 4.3 TRAINING STRATEGY

Our full loss is the sum of three complementary terms:

$$\mathcal{L} = \mathcal{L}_{cam} + \mathcal{L}_{depth} + \mathcal{L}_{pmap}.$$

We parameterise a camera by a unit quaternion $q \in \mathbb{R}^4$, a translation vector $trans \in \mathbb{R}^3$, and a shared focal length $f \in \mathbb{R}$.. The camera loss is a robust Huber metric, $\mathcal{L}_{\text{cam}} = \sum_{i=1}^{n} ||(\hat{g}_i - g_i)||_\epsilon$, comparing the ground truth $g_i$ and the predicted cameras $\hat{g}_i$. For every pixel $u$, the head outputs a depth estimate $\hat{D}_i(u)$ together with its positive uncertainty map (Kendall & Gal (2017); Novotny et al. (2017)). Hence, the depth loss is

$$L_{\text{depth}} = \sum_{i=1}^{N} || \sum_{i}^{D} \bigodot (\hat{D}_i - D_i)|| + || \sum_{i}^{D} \bigodot (\nabla \hat{D}_i - \nabla D_i)|| - \alpha log \sum_{i}^{D},$$

where $\bigodot$ is the channel-broadcast element-wise product. The point map loss is defined same but with the point-map uncertainty $\sum_{i}^{P}$:

$$L_{\text{pmap}} = \sum_{i=1}^{N} || \sum_{i}^{P} \bigodot (\hat{P}_i - P_i)|| + || \sum_{i}^{P} \bigodot (\nabla \hat{P}_i - \nabla P_i)|| - \alpha log \sum_{i}^{P}.$$

During the first stage of training, we focus exclusively on establishing robust intra-group alignment. To stabilize optimization and prevent the network from overfitting to short-range dependencies, we adopt a curriculum-style incremental schedule on the submap length. Specifically, we initialize training with very short submaps ($L_{\text{submap}} = 2$), and gradually increase $L_{\text{submap}}$ as training progresses. This progressive expansion encourages the model to adapt from local to increasingly long temporal horizons in a stable manner. During this training, we only open the weights of the final submap, facilitating a gradual training progression with larger increments.

In the second stage of training, we shift the optimization focus from intra-group refinement to inter-group alignment. To this end, the backbone parameters are frozen and only the TPV encoder and the cross-attention fusion modules are updated. To ensure stable convergence, we employ a zero-initialization strategy for the regression head, such that the initial transformation corresponds to an identity matrix. This design guarantees that the network starts from a well-posed alignment state, avoids introducing spurious distortions at the beginning of training, and facilitates stable optimization towards globally consistent reconstructions.

## 5 EXPERIMENTS

### 5.1 IMPLEMENTATION DETAILS

We use the weights of VGGT (Wang et al. (2025)) as pretrained weights. Our model is trained on two datasets: ScanNet (Dai et al. (2017)) and ScanNet++ (Yeshwanth et al. (2023)), which provide diverse 3D reconstructions of indoor environments, including RGB images and dense depth maps from various scenes. To validate our method, experiments are conducted on the 7-Scenes (Shotton et al. (2013)) and TUM RGB-D (Sturm et al. (2012)) datasets, both of which are real-world datasets consisting of partial scenes. The evaluation focuses on both dense mapping quality and camera pose estimation accuracy. Pose estimation accuracy is measured using Root Mean Square Error (RMSE) and Absolute Trajectory Error (ATE), while dense mapping performance is assessed through accuracy(the smallest Euclidean distance from the prediction to groundtruth) and completion(the smallest Euclidean distance from the ground truth to prediction) metrics (Grupp (2017)).

We configure the number of images per submap, $L_{\text{submap}}$, to 20 and define the number of submap per groups, $L_{\text{group}}$, 2. And number of overlap image $L_{\text{overlap}}$ set to 1. Employ the pointmap branch to evaluate the dense reconstruction performance. We set the image resolution to 640×480.

### 5.2 7-SCENES EVALUATION

For the 7-scenes dataset (Schonberger & Frahm (2016)), we use reported numbers from SLAM3R for baseline. We select one image every 15 frames. Both VGGT-SLAM (Wang et al. (2025)) and our method use a conference threshold of 3.0, where points with confidence scores below this threshold are filtered out, which follow the SLAM3R.

For reconstruction, we compare with Dust3R (Wang et al. (2024)), Mast3R (Leroy et al. (2024)), and Spann3R (Wang & Agapito (2024)) reconstruction approaches. Due to the VGGT-SLAM is the submap-based approch, we also report the results of VGGT-SLAM. As demonstrated in Table 1, our method achieves superior performance in both accuracy and completeness. Notably, the completeness of our approach significantly outperforms VGGT-SLAM . Our predictions, compared to projections, are better at capturing fine-grained details, thus effectively reducing errors.

Notably, on Office, RedKitchen, and Stairs, our method achieves the best completeness scores while maintaining competitive accuracy. These results highlight that our model is particularly effective at capturing fine-grained details and preserving scene structures, thereby reducing reconstruction errors arising from missing geometry.

The Root Mean Square Error (RMSE) of the Absolute Trajectory Error (ATE) on the 7-Scenes dataset is shown in Table 2. Add the SLAM-based approch NICER-SLAM (Zhu et al. (2024)) and DROID-SLAMTeed & Deng (2021). DROID-SLAM achieve the strongest overall performance. In certain scenarios, our method achieves better performance than VGGT-SLAM .

| Method | Chess Acc. /Comp. | Fire Acc. /Comp. | Heads Acc. / Comp. | Office Acc. / Comp. | Pumpkin Acc. / Comp. | RedKitchen Acc. / Comp. | Stairs Acc. / Comp. | Avg. Acc. / Comp. |
|---|---|---|---|---|---|---|---|---|
| DUSt3R | 2.26 / 2.13 | 1.04 / 1.50 | 1.66 / **0.98** | 4.62 / 4.74 | 1.73 / 2.43 | 1.95 / 2.36 | 3.37 / 10.75 | 2.19 / 3.24 |
| MASt3R | 2.08 / 2.12 | 1.54 / 1.43 | **1.06** / 1.04 | 3.23 / 3.19 | 5.68 / 3.07 | 3.50 / 3.37 | 2.36 / 13.16 | 3.04 / 3.90 |
| Spann3R | 2.23 / 1.68 | 0.88 / 0.92 | 2.67 / **0.98** | 5.86 / 3.51 | 2.25 / **1.85** | 2.68 / 1.80 | 5.65 / 5.15 | 3.42 / 2.41 |
| SLAM3R | **1.63 / 1.31** | **0.84 / 0.83** | 2.95 / 1.22 | **2.32** / 2.26 | 1.81 / 2.05 | 1.84 / 1.94 | 4.19 / 6.91 | 2.13 / 2.34 |
| VGGT-SLAM | 2.06/ 3.67 | 1.38 / 2.20 | 2.13 / 2.60 | 2.68 / 4.87 | **1.66** / 2.47 | 2.69 / 4.09 | 1.91 / 2.23 | 2.07 / 3.16 |
| Ours | 2.21 / 4.78 | 2.00 / 1.62 | 1.53/ 1.05 | 2.68 / **1.68** | 2.39 / 1.93 | **1.59 / 1.76** | **1.61 / 1.86** | **2.00 / 2.10** |

Table 1: Reconstruction results on 7 Scenes dataset(unit: cm). The **bolded** values represent the best results, and the underlined values represent the second-best. Lower Acc. and Comp. indicate better camera pose estimation

### 5.3 TUM RGB-D EVALUATION

We evaluate DROID-SLAM, MASt3R-SLAM in Tum RGB-D. Although our method does not achieve the highest average performance, it demonstrates superior accuracy in pose estimation in certain scenarios. As shown in Table 3, while our method exhibits a relatively low Root Mean Square Error (RMSE) in some scenes such as Room and XYZ. This result suggests that our method

| Method | Scenes | | | | | | | Avg. |
|---|---|---|---|---|---|---|---|---|
| | Chess | Fire | Heads | office | Pumpkin | RedKitchen | Stairs | |
| **DUSt3R** | 0.050 | 0.048 | 0.025 | 0.012 | **0.010** | **0.010** | **0.010** | 0.080 |
| **MASt3R** | 0.043 | 0.029 | **0.014** | 0.012 | 0.011 | 0.079 | 0.030 | 0.062 |
| **NICER-SLAM** | **0.032** | 0.068 | 0.041 | **0.010** | 0.020 | 0.039 | **0.010** | 0.085 |
| **DROID-SLAM** | 0.033 | **0.024** | **0.014** | 0.091 | 0.016 | 0.049 | 0.018 | **0.056** |
| **Spann3R** | 0.091 | 0.066 | 0.071 | 0.215 | 0.128 | 0.140 | **0.140** | 0.117 |
| **SLAM3R** | 0.062 | 0.053 | 0.045 | 0.124 | 0.117 | 0.094 | 0.092 | 0.084 |
| **VGGT-SLAM** | 0.036 | 0.028 | 0.018 | 0.103 | 0.133 | 0.058 | 0.093 | 0.067 |
| **Our** | 0.061 | 0.073 | 0.020 | 0.093 | 0.110 | 0.077 | 0.087 | 0.076 |

Table 2: Root Mean Square Error (RMSE) of Absolute Trajectory Error (ATE) on 7-Scenes dataset (unit: m). The **bolded** values represent the best results, and the underlined values represent the second-best. Lower values indicate better camera pose estimation.

excels in specific environments, potentially due to its ability to capture finer scene details or handle particular geometric properties better.

| Method | Scenes | | | | | | | | | Avg. |
|---|---|---|---|---|---|---|---|---|---|---|
| | 360 | Desk | Desk2 | Floor | Plant | Room | RPY | Teddy | XYZ | |
| **DROID-SLAM** | 0.202 | 0.032 | 0.091 | 0.064 | 0.045 | 0.918 | 0.056 | 0.045 | **0.012** | 0.158 |
| **MASt3R-SLAM** | **0.070** | 0.035 | 0.055 | **0.056** | 0.035 | 0.118 | 0.041 | 0.114 | 0.020 | 0.060 |
| **VGGT-SLAM** | 0.071 | **0.025** | **0.040** | 0.141 | **0.023** | 0.102 | 0.030 | **0.034** | 0.014 | **0.053** |
| **Our** | 0.124 | 0.031 | 0.089 | 0.102 | 0.025 | **0.100** | 0.040 | 0.042 | **0.012** | 0.062 |

Table 3: Root mean square error (RMSE) of absolute trajectory error (ATE) on TUM RGB-D dataset (unit: m). The **bolded** values represent the best results, and the underlined values represent the second-best. Lower values indicate better camera pose estimation.

## 5.4 Ablations

We test the inference efficiency on an NVIDIA H100 GPU, with all $L_{\text{group}}$ set to 2 and $L_{\text{submap}}$ set to 21 (with an overlap frame).We compare the runtime with VGGT (Wang et al. (2025)), and our method.

We evaluate runtime performance by comparing VGGT with our method, with and without the spatial memory $\mathcal{M}$ for inter-group alignment.

The results in Table4 show that our method achieves a significant speedup over VGGT. Moreover, the spatial memory introduces only negligible overhead, indicating that our approach preserves efficiency while improving consistency. When processing 120 frames, our method reduces the runtime from 8.40s to 2.90s, corresponding to a ∼3× improvement.

We further evaluate the effect of incorporating the spatial memory. As shown in Table 5, leveraging spatial memory improves both accuracy and completeness, while maintaining the performance of camera pose estimation.

| Method | 60 | 80 | 100 | 120 |
|---|---|---|---|---|
| VGGT | 3.56 | 3.73 | 5.87 | 8.40 |
| Our(w/o $\mathcal{M}$) | 1.42 | 1.89 | 2.38 | 2.82 |
| Our(W/ $\mathcal{M}$) | 1.69 | 1.96 | 2.4 | 2.90 |

Table 4: Ablation study on inference efficiency.

| Method | Recon. | | Camera. |
|---|---|---|---|
| | Acc. | Comp. | RMSE |
| Our(w/o $\mathcal{M}$) | 2.027 | 2.135 | 0.076 |
| Our(W/ $\mathcal{M}$) | 2.007 | 2.101 | 0.076 |

Table 5: Ablation study of reconstruction results (cm) and Root Mean Square Error (RMSE) of Absolute Trajectory Error (ATE) (m) on the 7-Scenes dataset.

## 5.5 Qualitative Analysis

We selected scenes from both the TUM RGB-D (Sturm et al. (2012)) and 7-Scenes (Schonberger & Frahm (2016)) datasets and used COLMAP (Schonberger & Frahm (2016)) to reconstruct them as ground truth.

As shown in Figure 3, in the first scene, we successfully reconstructed the stair, whereas VGGT-SLAM (Wang et al. (2025)) exhibited misalignment, and SLAM3R failed to produce a valid reconstruction. Our method demonstrated a more accurate reconstruction the geometry of stair.

The second scene is a typical example of a small-scale, complex environment featuring multiple orthogonal walls, a tabletop, and various cluttered items. VGGT-SLAM suffers from layering artifacts when there is a significant discrepancy in the predicted scales between consecutive frames. In contrast, our model effectively mitigated the wall separation issue, achieving a consistent reconstruction across the entire plane. Accurate scale prediction is crucial for this scenario. Both SLAM3R and VGGT-SLAM failed to accurately reconstruct the walls, resulting in layer separations. . In contrast, our model effectively mitigated the wall separation issue.

These scenes highlight the capability of our network to effectively capture and learn the scale of spatial details.

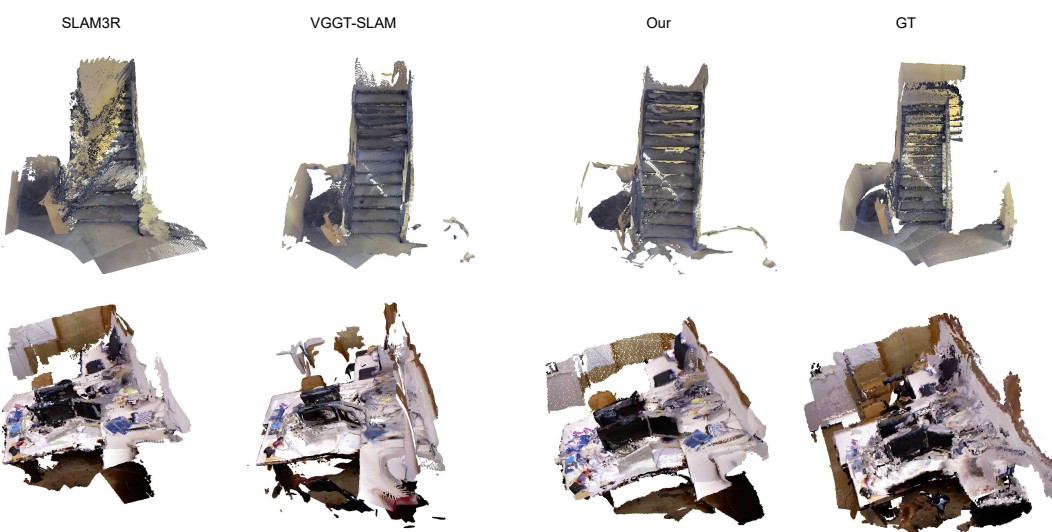

Figure 3: Qualitative reconstruction results on two representative indoor scenes: the Stairs sequence from the 7-Scenes dataset and the Desk sequence from the TUM RGB-D dataset. Our method produces more faithful and complete reconstructions compared to existing baselines.

## 6 LIMITATIONS

Although ES-GGT delivers competitive trajectory ATE in most indoor scenes, its camera poses still lag behind some SLAM systems such as DROID-SLAM (Teed & Deng (2021)) and VGGT-SLAM (Maggio et al. (2025)) (Table2 & 3). The gap is most evident in rapid-rotation or texture-poor sequences the TPV memory provides only weak metric anchoring. To bridge the gap in pose accuracy, we need to devise a more effective alignment strategy, which leading to smaller inter-group errors.

## 7 CONCLUSION

We presented ES-GGT, an architecture build on VGGT (Wang et al. (2025)) that enables efficient 3D reconstruction from monocular RGB images. Our method achieves superior reconstruction accuracy and completeness on 7-scenes dataset, and a significant speedup over VGGT.

By combining local refinement with global spatial memory, ES-GGT achieves both accuracy and efficiency, paving the way for practical long-horizon 3D reconstruction. Experiments demonstrate the effectiveness of our local-to-global strategy.

# 8 ETHICS STATEMENT

We employed large language models solely for language editing and translation of the manuscript. No part of the method design, experiments, or analysis relied on LLM-generated content.

# 9 REPRODUCIBILITY STATEMENT

The source code to reproduce the main results will be released upon publication.

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
