# OpenReview forum: "ES-GGT: Efficient Submap-based Visual Geometry Grounded Transformer with Spatial Memory Alignment"
_ICLR.cc/2026/Conference — Submitted to ICLR 2026_

### Official Review · Reviewer_ur6a · 2025-10-27

**Soundness:** 2
**Presentation:** 1
**Contribution:** 2
**Rating:** 2
**Confidence:** 5

**Summary:**

This paper proposes a method for streaming 3D reconstruction based on VGGT to handle longer sequences and improve the inference speed.
It divides the input images into groups and submaps, processes each submap separately, and conducts local and global alignment to ensure the overall consistency.

**Strengths:**

The problem this paper wants to solve is meaningful and significant (the efficiency when processing long sequences).

**Weaknesses:**

1. From my understanding, in each submap, the predicted pointmaps and poses share the same coordinate system. When conducting the local alignment intra each group, ES-GGT uses a lightweight MLP to align the camera tokens. Why is it unnecessary to design a similar alignment mechanism for the image tokens here? Since both the camera tokens and image tokens of the same image are obtained together from the backbone, would applying such a direct adjustment only to the camera tokens cause inconsistency in the output?
Besides, Figure 2 has some mistakes. In the intra-group alignment module, it should be 'Submap1', 'Submap2', ..., not 'Group1', 'Group2'.
As groups and submaps are different concepts in this paper, this mistake will cause confusion.
2. I think Section 4.2 is hard to read. Given a new group output, since the global alignment proposed in this section is intended to align different groups, why is the point cloud first preprocessed using the Sim(3) method? And how is M queried? I believe these details are essential for understanding this module, but they are not clearly explained here.
3. It seems that ES-GGT only uses two datasets to train. I want to know the training costs (time, GPUs).
4. What is the input resolution in Table 1's comparison? ES-GGT is trained with a resolution of 480*640. Up to my knowledge, Spann3R is trained with a resolution of 224. I think the resolution used to evaluate is important as this pipeline is not resolution-adaptive.
5. I think you should compare the reconstruction performance of ES-GGT with CUT3R and VGGT. And please provide the results on NRGBD datasets. Now the results (both reconstruction and pose) can not prove the effectiveness of ES-GGT.
6. If the speed when processing long sequences is the core advantage, please provide the running time of each module in your pipeline (compared with VGGT), and thus analyze the efficiency. And reconstruction results on longer sequences can make your statement more faithful and solid.

Minor Weaknesses:
1. There seems to have missing content in Line 72.
2. Some typos and some formats are wrong (like 'unmber' in Line 75, Line 257-259, and so on).

By the way, I cannot open the Code link.
Please answer the above questions carefully and provide more thorough discussion and comparison.

**Questions:**

Please refer to the Weaknesses.

---

### Official Review · Reviewer_4ehA · 2025-10-31

**Soundness:** 3
**Presentation:** 2
**Contribution:** 2
**Rating:** 6
**Confidence:** 4

**Summary:**

ES-GGT presents an efficient, online approach to monocular RGB 3D reconstruction by extending VGGT with a hierarchical, submap-based streaming framework that reduces computational complexity from $\mathcal{O}(N^2)$ to $\mathcal{O}(N \cdot L_s)$, where $N$ is the total number of frames and $L_s$ is the submap size. Its key innovations include an intra-group alignment mechanism that enforces local geometric consistency by propagating overlap-frame camera features through shared attention layers, and an inter-group alignment module that uses a Tri-Perspective View (TPV) spatial memory and a 15-DoF rigid transformation head to globally align group outputs without iterative optimization. This enables end-to-end, single-pass reconstruction of long sequences while preserving geometric interpretability. Experiments on the 7-Scenes dataset demonstrate state-of-the-art reconstruction accuracy and completeness, with a nearly 3× speedup over VGGT for sequences of 120 frames, making it a practical solution for real-time, large-scale 3D scene modeling.

**Strengths:**

S1: ES-GGT innovatively fuses streaming submaps with a TPV-based spatial memory to achieve end-to-end geometric alignment without costly optimization loops.

S2: The experiments are thorough—demonstrating over 3× faster inference while achieving state-of-the-art completeness on 7-Scenes.

S3: Diagrams and explanations are exceptionally clear, making the two-stage alignment mechanism intuitive even to non-specialists.

S4: It transforms long-sequence 3D reconstruction from a theoretical challenge into a practical, real-time capability for robotics and AR applications.

**Weaknesses:**

W1: TPV memory uses fixed voxel grids; adaptive resolution or dynamic memory pruning could reduce memory overhead for large scenes.

W2: Evaluation lacks results on texture-poor or rapid-motion sequences, where SLAM systems like DROID-SLAM excel—missing a key stress test.

W3: No ablation on $L_{\text{overlap}}$ or $L_{\text{group}}$ sensitivity; optimal settings appear tuned for 7-Scenes without generalization analysis.

W4: Code is anonymized and not yet public—hinders reproducibility and fair comparison with recent baselines like MapAnything or Fast3R.

**Questions:**

See weakness

---

### Official Review · Reviewer_cqbD · 2025-10-31

**Soundness:** 3
**Presentation:** 2
**Contribution:** 2
**Rating:** 6
**Confidence:** 4

**Summary:**

1 The paper proposes ES-GGT, an efficient submap-based extension of VGGT that reduces computational complexity via a hierarchical local-to-global alignment strategy using overlapping submaps and spatial memory with Tri-Perspective View (TPV).

2 It evaluates performance on 7-Scenes and TUM RGB-D datasets using reconstruction metrics (accuracy/completeness) and pose estimation (ATE RMSE), showing superior completeness and ~3× speedup over VGGT on long sequences (>100 frames).

3 Ablation studies confirm the effectiveness of intra-group alignment and the minimal overhead of spatial memory, though camera pose accuracy still lags behind top SLAM methods like DROID-SLAM in challenging scenes.

**Strengths:**

The paper presents a well-motivated and technically sound approach that creatively combines submap-based processing, hierarchical alignment, and spatial memory with TPV representation to address the scalability limitations of VGGT. Its originality lies in the efficient integration of intra-group fine-grained consistency and inter-group global alignment within a single feed-forward framework, avoiding costly post-hoc optimization. The method is clearly described, supported by comprehensive experiments on standard benchmarks, and demonstrates meaningful improvements in both reconstruction completeness and inference speed—particularly on long sequences—making it a significant contribution toward practical, large-scale 3D reconstruction from monocular video.

**Weaknesses:**

1 The grouping strategy for submaps is critical yet under-specified—grouping spatially adjacent vs. random frames may significantly affect reconstruction consistency, especially in large-scale scenes, but the paper lacks analysis on how grouping choices impact performance.

2 The experiments compare against VGGT-SLAM but omit baseline VGGT in Tables 1–3, and Table 4’s runtime comparison excludes VGGT-SLAM, making it unclear how ES-GGT truly stacks up against both the original and SLAM-adapted versions in accuracy and speed.

3 The paper provides minimal qualitative results—missing camera trajectory visualizations and diverse scene comparisons—which weakens confidence in the claimed improvements, as the few shown examples are insufficient to demonstrate robustness or generalization.

**Questions:**

Similar to Weakness.

---

### Official Review · Reviewer_iTK5 · 2025-11-03

**Soundness:** 3
**Presentation:** 2
**Contribution:** 2
**Rating:** 4
**Confidence:** 4

**Summary:**

This paper proposes ES-GGT, an efficient transformer-based approach for dense 3D scene reconstruction from monocular RGB sequences. ES-GGT extends the VGGT model by introducing a submap-based, hierarchical local-to-global alignment strategy, leveraging intra-group fine-grained alignment and inter-group spatial memory via Tri-Perspective View (TPV) representations.Experiments on standard indoor benchmarks demonstrate significant improvements in computational efficiency and reconstruction quality compared to prior methods including VGGT and its SLAM variants.

**Strengths:**

Strengths:
1. The paper's structure is clear and easy to read.
2. The performance of ES-GGT looks good and improve the efficiency.

**Weaknesses:**

Major Weakness:
1. The evaluation omits the simplest baselines, e.g. VGGT, since ES-GGT use the weight and model of VGGT
2. The paper claims to avoid projective ambiguity by replacing SL(4) homography optimization with a learnable 15‑DoF homography matrix. However, the motivation for a 15‑DoF homography in this context is not clearly derivedThe 15‑DoF transformation is introduced without a clear derivation. Readers may wonder why fifteen degrees of freedom are needed and how this relates to SL(4) or Sim(3)

Minor Weakness:
1. in line 414, the cross reference format is inconsistent. In line 480, build should be built

**Questions:**

Pleaes see the weakness above.
Also:
1.The 15‑DoF transformation is introduced without a clear derivation. I wonder why fifteen degrees of freedom are needed and how this relates to SL(4) or Sim(3)
2. The progressive submap length training and two‑stage freezing of the backbone are reasonable. Yet it would be helpful to show ablations demonstrating why these strategies are necessary (e.g., performance curves vs. submap length).

---

### Meta-Review · Area_Chair_Uscd · 2025-12-31

**Summary:**

Despite mixed original reviewer scores, reviewers raised many important concerns. These include a lack of comparison to some other methods, including the VGGT as a baseline since the proposed method is directly derived from it, limitation experimental evaluations, problems with readability, including an incomplete paragraph at Line 072.

**Reviewer Concerns:**

There was no rebuttal, and no paper revision. Hence all concerns remain outstanding.

**Reviewer Scores:**

Despite the original reviewer scores being mixed, the lack of rebuttal comments or paper revision / corrections would likely mean reviewers downgrading their scores. The AC does not consider the current submission, without extensive revision based on reviewer requests and questions, to be in shape for publication.

---

### Decision · Program_Chairs · 2026-01-26

Reject